# The Role of Natural and Human Resources on Economic Growth and Regional Development: With Discussion of Open Innovation Dynamics

**Haeruddin Saleh [1],\*** , **Batara Surya [2]** , **Despry Nur Annisa Ahmad [3] and Darmawati Manda [4]**

[1]  Regional Economic Department, Faculty of Economics and Business, University Bosowa,
    Makassar 90231, Indonesia
[2]  Department of Urban and Regional Planning, Faculty of Engineering, University Bosowa,
    Makassar 90231, Indonesia; batara.surya@universitasbosowa.ac.id
[3]  Department of Urban Regional Planning, Faculty of Science and Technology Alauddin State Islamic
    University, Makassar 92133, Indonesia; despry.nur@uin-alauddin.ac.id
[4]  Department of Financial Economics, Faculty of Economics and Businness University Bosowa,
    Makassar 90231, Indonesia; darmawati.manda@universitasbosowa.ac.id
**\***  Correspondence: haeruddin@universitasbosowa.ac.id

**Abstract:** Regional economic growth is closely related to the optimization of the use of natural and human resources. This study aims to analyze: (1) The use of natural and human resource potential works as a determinant of the economic growth of Bulukumba Regency; (2) The influence of natural resources, human resources, community culture and regulation on economic growth in Bulukumba Regency. The research method used is a combination of mixed models, namely a combination of quantitative and qualitative approaches. Data was obtained through observation, survey and documentation. The results of the study show that optimizing the use of natural resources without human resource development causes its contribution to economic growth in Bulukumba Regency to be quite low which becomes an obstacle to the acceleration of economic development. The influence of natural resources, human resources, and community culture together has a significant effect on economic growth in Bulukumba Regency with a coefficient of determination 47.2%. This study recommends optimizing resource potential and strengthening human resource capacity through the use of technology and changes in community culture, which will encourage economic growth in Bulukumba Regency, South Sulawesi Province, Indonesia.

**Keywords:** economic growth; natural resources; human resources; utilization of technology

## 1. Introduction

A country is said to be advanced when its economic growth is higher than in other countries and it has a GDP per capita of 12,375 USD [1]. One of the indicators of an advanced country is economic growth 7.5% off from the state trap with a middle-income trap [2]. China, in 2010, experienced rapid economic growth, reaching 10% [3]. The growth was due to increased capital accumulation affected by tariff magnitude. Reductions in the tariffs on capital goods and intermediate inputs led to higher investment in foreign capital goods, whereas reduction in the output tariff resulted in lower investment [4]. A lot of capital accumulation in a country will drive increased productivity of products and services. Successful catchup requires the ability to produce goods of better quality and lower prices than those produced by incumbent firms from advanced countries [5].

The availability of natural resources in a country with high economic value suggests, in theory, that its economy will be easier to develop. However, resource abundance often causes distortions or

certain tendencies in an economy, and these distortions then undermine economic performance [6–8] in the context of a developed country with a high GDP per capita such as Japan (48,920 USD), Netherlands (55,041 USD) and Singapore (58,248 USD) [9]. The economic development of these countries does not depend on their natural resources factor, but rather relies on the ability of their human resources. Resource-abundant countries tend to be high-price economies and, perhaps as a consequence, these countries tend to miss-out on export-led growth [10,11]. The human component in sustainable development plays a major role [12]. The high quality of the workforce, through increased knowledge and mastery of technology in a country, can increase productivity. When discussing the growth of either developed countries or countries in general, factors like education and technological improvements have a positive correlation with economic growth [13,14].

Indonesia is an archipelago with a diversity of natural resources, and its economic growth only reaches 5.3%. The economic growth of other Southeast Asian countries such as the Philippines (6.52%), Vietnam (6.60%) and Cambodia (6.90%) has developed due to increased investment from various sectors, especially the manufacturing sector [15]. The mechanism of the innovation multiplier is based on the fact that an additional change in investments in the innovations of some business units becomes the income of other economic entities [16]. Indonesia's economic growth could reach 6% if various policies are supported, regulations are conducted that do not support investors and bureaucracy is immediately removed in support of export activities. Bureaucratic violence is fundamental to the generation of value in the green economy, as a process that works alongside the commodification spectacle, and other forms of structural and material violence [17].

Another policy factor undertaken by the Indonesian government in supporting economic growth is the implementation of fiscal policy. Fiscal policy has a significant regulatory impact on the economic processes through an integrated combination of fiscal architectonics instruments [18,19]. Taxation and expenditure are the instruments used in the implementation of fiscal policy. Improvement in government expenditure on health, education and economic services, as components of productive expenditure, boosts economic growth [20–22]. Many sources of financing for the implementation of development come from taxation and high aggregate expenditures, which can encourage regional economic growth. Investment expenditures, in particular investment in human resources, should be prioritized higher than all other items of spending, assuming economic growth [20,23]. This, in turn, benefits the entire economy, whereas primary production does not require high levels of human capital [24,25].

Bulukumba Regency is an area rich in natural resources, but these resources have not contributed much to the regional economic growth. From the 2019 data, the economic growth of Bulukumba Regency is 5.05%. Other areas in South Sulawesi Province include Bantaeng Regency (8.08%), Bone District (8.90%), and Luwu Regency (8.42%) [26]. The low economic growth of Bulukumba regency is due to the lack of investment activities that process the existing natural resources. Bulukumba Regency Government seeks to support the acceleration of economic growth through policies focused on improving economic stability, socio-politics, security, and law enforcement to encourage investment. Policy interventions are used to assist emerging economic firms in building their absorptive capacity and strengthening their learning intent in order to promote innovation and improve their value-added position [27–29]. Such firms face particular difficulties in successfully pursuing innovative strategies, which may perhaps be reduced in cities and regions where agglomeration economies or open networks allow more of the support facilities and uncertainties to be externalized [30]. The economic growth that remains in other regions is a fundamental problem, so strategies are needed to advance again. Th objectives of this study are (1) to determine the potential use of natural resources and human resources as a determinant of economic growth in the Bulukumba district and (2) to determine the influence of natural resources, human resources, community culture and regulation on economic growth in Bulukumba district.

## 2. Conceptual Framework for Economic Growth in Bulukumba Regency

Development of the region to achieve economic growth should be done by processing natural resources optimally and sustainably, resulting in the competitiveness of products from the region. Planned investment in a new mine level and additional pellet capacity, along with continued product development efforts, assure a company's long-run viability [31]. The level of competitiveness is one of the parameters in the concept of sustainable economic development, so it needs to be supported by a clear vision. Competitiveness should be underpinned by a broad vision for the economy and society [32]. From a macro perspective, the prosperity of a region in terms of economic performance depends on the ability of the region to create job opportunities and increase the real income of its inhabitants. Economic prosperity is very dependent on the productivity of the population of a nation [33,34]. Productivity is seen as the main determinant in the long term for the rise of living standards in regions such as the Bulukumba district.

A strategy is needed to develop the stale sector in Bulukumba Regency. The quality of the formed strategy of regional development depends on the accounting of the interests of the state, business and inhabitants as the basic elements of the regional social and economic system. Such an approach allows us to increase the quality of regional strategies for development and account for the complexities of territory development planning in general [35]. The competitiveness of the economy and industry is influenced by internal factors and external factors [36]. Natural resources are an internal factor and government policy is an external factor, and both decide the sustainable economic growth of the region. Whatever the chosen development trajectory and policy regime, one important lesson is that they are unlikely to be effective and sustainable without a full appreciation of the trans-local dynamics in which the region is located. This is the key contribution of regional development, as necessarily situated in the competitive dynamics of global production networks [37]. The estuary of policy implementation is the achievement of the productivity of a region. Policy implementation at the operational and service levels also involves coordination with other organizations, including those that may have no previous experience working together, which may have either a positive or negative effect on service [38,39].

High competitiveness can encourage increased investment in an area. Investment is a key determinant of the rate of economic growth because in addition to pushing for a significant increase in the output it will also automatically increase demand for inputs, which in turn will increase employment opportunities and public welfare as a consequence of increased income received by the community [40]. Thus, the development of investment in a region especially in Bulukumba Regency has many benefits and economic growth is increasingly advanced. It also finds that higher life expectancy and increases in investment have a positive impact on economic growth [41]. The Problem of Investment appeal assessment has a sufficient theoretical basis. Investment in conjunction with regional development issues is discussed [42–44]. For more details about the Conceptual Framework, see Figure 1 as follows:

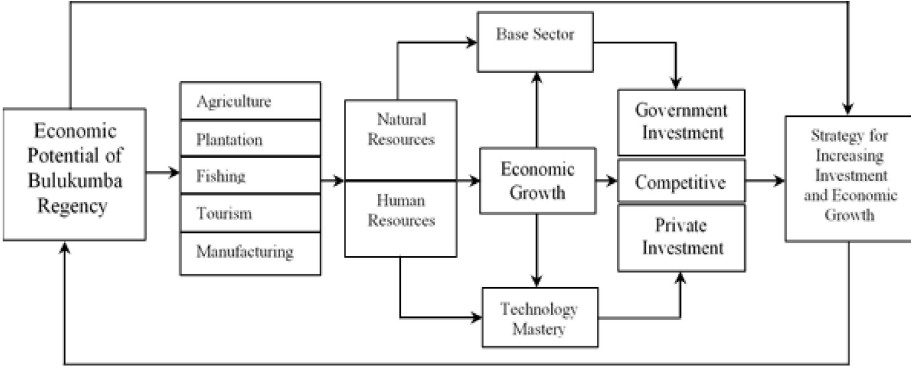

**Figure 1.** Conceptual Framework for Economic Growth in Bulukumba Regency. Source: Author Elaboration.

### 2.1. Regional Characteristics and Potential

The location of Bulukumba Regency in the south of the South Sulawesi region with a distance of ±163 km from the capital of South Sulawesi Province is located between 050 20′–050 40′ South Latitude and 1190 58′–1200 28′ East Longitude. Bulukumba Regency has a large area which reaches 1145.67 km$^2$ or 2.5% of the total area of South Sulawesi. Bulukumba Regency has ten sub-districts consisting of 27 villages with a total number of villages reaching 109. Bulukumpa Regency has a varied morphology with a slope of land from 0–400, while the height of the sea level reaches 0–1000 m or 95.4%. Regarding the potential of the area in Bulukumba Regency, which consists of sloping areas (0–3%), this area occupies approximately 45 km$^2$ or 3.3% of the total area of the regency, which is a cultivation area of paddy fields, plantations, settlements and aquaculture. The slope area (3–5%) occupies 135 km$^2$ or 11.8% of the area of Bulukumba Regency; it is an area of cultivation, rice fields, settlements and other public facilities. Slope area 5–10% occupies 250 km$^2$ or 21.7 percent and constitutes plantation areas and limited production forest areas. Slope area 10–15%, occupying 325 km$^2$ or 28.3% of the area of Bulukumba Regency, is a plain area that is partly a limited production forest area and partly a plantation area. Slope area 15–30% occupies 210 km$^2$ or 18.3% of Bulukumba Regency area; most of this area is a limited production forest area, a plantation cultivation area and a settlement. Slope area 30–70% occupies 169.7 km$^2$ or 14.8%; most of this area is limited production forest and protected debt. Slope area >70% occupies approximately 20 km$^2$ or 1.8% of the area of Bulukumba Regency, which is a mountainous area and protected forest area [29].

Lowland areas with altitudes between 0 to 25 m above sea level include seven coastal districts, namely Gantarang District, Ujung Bulu District, Ujung Loe District, Bontobahari District, Bontotiro District, Kajang District and Herlang District. The corrugated area with a height between 25 to 100 m above sea level includes parts of Gantarang District, Kindang District, Bontobahari District, Bontotiro District, Kajang District, Herlang District, Bulukumpa District, and Rilau Ale District. The hilly area in Bulukumba Regency extends from West to North with an altitude of 100 to 500 m above sea level covering parts of Kindang District, Bulukumpa District and Rilau Ale District [29]. For more details, the map of potential areas can be seen in Figure 2 as follows:

The economic potential of Bulukumba Regency can be developed in accordance with the regional development framework. The study proves that investors, when making decisions about investing their funds, pay attention primarily to the extent of intellectual potential development in a region, rather than to labor availability. Consequently, the strong influence of this factor on the inflow of investment makes it possible to conclude that regions that have reached the greatest level of intellectual potential development have every reason to be the most attractive for investors [45]. The notion and characteristics of economic growth through the prism of cyclical fluctuations in economic systems depending on the level and rate of the development of entrepreneurship have been found [46–49]. Minapolitan development, agropolitan areas, agro-tourism areas, tourism development centers, trade areas, passenger airport areas and port areas are the driving forces for economic growth. Innovative research, mixing network analysis and strategic investment analysis, like most works in this domain, remain disconnected from the 'ground' and mostly focus on topological properties (in physics, for instance). Maritime networks are spatial networks and therefore evolve according to multiple complex factors including decision making, the specific characteristics of the connected nodes and the regions hosting them [50].

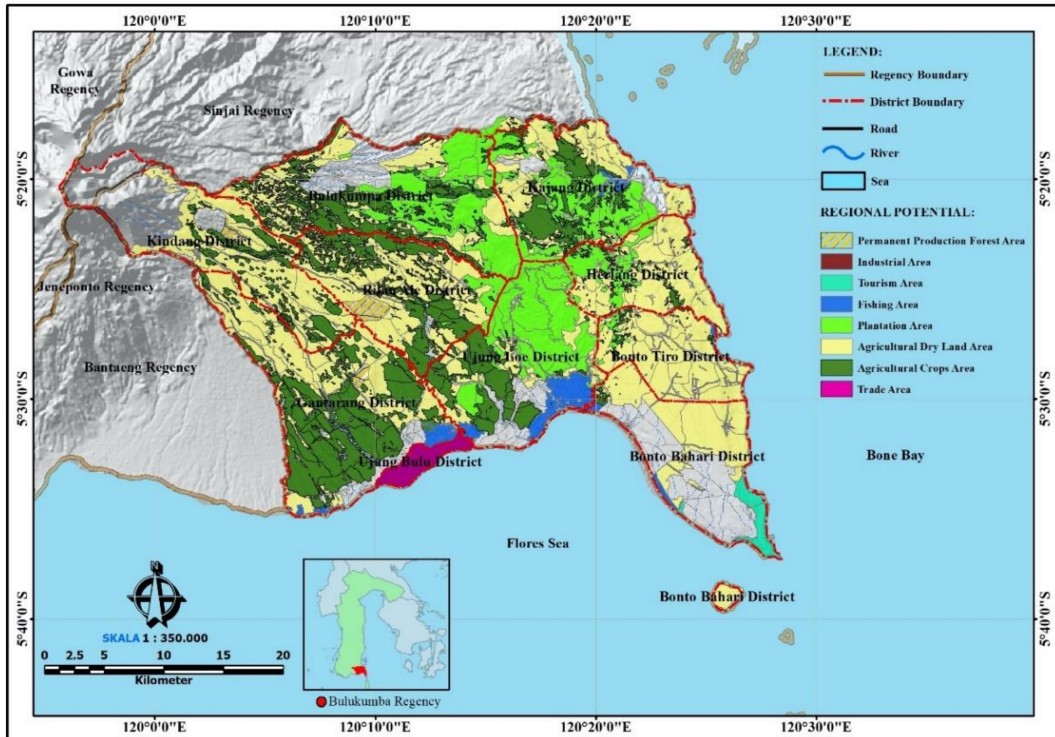

**Figure 2.** Administrative map and regional potential. Source: Authors; Map© Google

## 2.2. Investment Opportunities in Bulukumba Regency

Investment can be formed, greatly influenced by the role of the community in the use of funds, when income owned is not fully used for consumption but is instead put in the tube. Early evidence suggests that local government spending, in particular where the capacity of local government is limited, can crowd out private sector activities through public sector burgeoning [51].

Investment opportunities are determined by the internal and external aspects of the production process of a business. Findings indicate that international experience, product technology, degree of internationalization, market share and certain external factors influence the weighting managers give to internal and external factors in the process of making international pricing decisions [52]. Internally within a company quality human resources and technological knowledge is needed to manage the resources owned in an area. The external aspect is the macroeconomic condition, both in the social and political economy. Government policies can influence the decision to invest in an area, if the tax rate is high, and they can also reduce interest in investing in an area. We find the incentives to invest in foreign operations are primarily driven by tax incentives, government agencies and formal sector companies, and they can be easily created with capacity building improvement schemes. However, these agencies and companies cannot compete with businesses with larger companies and thus there is a need for governments to accelerate their growth by creating taxation, creating an environment for them via appropriate tax incentives that enhances their sustainability and growth [53,54].

Bulukumba Regency has the potential to be developed in the fields of agriculture, plantations, animal husbandry, fisheries, mining and energy which are supported by regional tourism development. The economic benefits of resource extraction are felt by those living close to resources [51]. From the data available in 2018 the agricultural sector made a large contribution to the economic growth of Bulukumba Regency reaching 40.9%, but this sector has decreased from year to year. Despite previous economic growth, the world is headed toward resource depletion and serious social crises, and old ways of problem-solving have proven inadequate. Something has to be done to change development—its philosophy and methods—if societies want to reverse those negative trends [55]. The expected role of

investors is to invest in the sector because it has great potential. While the mining sector is still a very small contributor at only 2.65%, the growth rate from year to year has experienced a very large increase reaching 14.89%. Other sectors that have a large contribution include the trade sector, reaching 15.89%, while other sectors are still very small contributors, so the opportunity for investment in this region is very large.

Investment appeal is a generalized characteristic of a set of social, economic, organizational, political and socio-cultural preconditions that determine the attractiveness and expediency of investing in one or another economic system [56,57]. In many developing countries social systems can act as either drivers of or constraints on economic development. Obstacles can affect if, in an area having a culture that does not keep up with the times (such as the use of technology), the community refuses to use more productive and more efficient methods or means of production. To understand the social foundation of technology from the perspective of economic history we need to study how historians view economic growth. History reveals that the weakening and disappearance of religion undermined social norms and values upon which economic behavior, organization and control depend. We know that because economies that retained and were able to improve their traditional institutions have done well relative to their counterparts who lost all or most of their traditional institutions and had to start from scratch [58].

## 3. Material and Methods

This research is a type of qualitative survey research that describes systematically, factually and accurately a treatment in the area that is the object of research, while the quantitative method for testing the hypothesis consists of:

**Hypothesis 1 (H1).** *The economic potential of Bulukumba District can increase the economic growth of the region.*

**Hypothesis 2 (H2).** *The influence of natural resources, human resource competencies, culture and regulatory factors in the region of Bulukumba Regency can encourage economic growth with strategies according to regional characteristics.*

This is done to reveal trends and to prove simple statistics using various quantitative data. The philosophical reason for combining both approaches was triangulation logic, and qualitative research results were re-checked in the quantitative approach and vice versa [59].

### 3.1. Data Collection Methods and Techniques

The study used a combination of mixed models, namely a combination of quantitative and qualitative approaches. Meanwhile, data collection techniques were obtained through observation, survey and documentation. Observations were made to record conditions in the field, characteristics and potential of the area and investment opportunities. For observations in this study, we used field notes, periodical notes and checklists of regional potentials. The data collected through observation were the number of farmers, the number of those who carry out agricultural activities of food crops, the number of farmers who carry out plantations, the number of farmers who carry out animal husbandry, the number of farmers in the fisheries sector and the number of people who carry out activities other than in the agricultural sector. In addition, we collected data on how the existing culture in the community is related to agricultural business activities and other economic activities. The aim was to find out the situation and potential areas associated with research. Survey data is the collection of data at certain times with the aim of (1). describe the natural state that exists now, (2). identify and measure the current state for comparison, and (3). determine the relationship between the parts. The documentation in this study includes population data obtained from the Statistics Office of Bulukumba Regency, the amount of land and superior production of agricultural products, the amount

of production of products outside of agricultural products, data on the amount of income per capita and data on reports on regulations related to the agricultural, plantation, fishery sectors, industry and tourism. The data collected through observation, surveys, in-depth interviews and documentation are then categorized as primary data, including data obtained directly by researchers in the field. Data obtained through surveys are categorized as quantitative data, while data obtained through in-depth interviews and observations are categorized as qualitative data.

### 3.2. Population and Sample

Population research relates to entire groups of people, events, or objects as the focus of research [60]. The population in this study is the total number of reports of data sources from respondents concerning economic actors, while the sample is part of the number and characteristics possessed by the population. If a population is too large and researchers are not able to study everything in the population, then a sample is used, which is part of the population to be studied and which is considered to be able to describe the population. The research sample is an important factor that needs attention in our research. The research sample reflects and determines how useful the sample is in making research conclusions. A small sample of population groups is taken according to the procedure so that they can represent the population. Sampling of respondents was based on purposive sampling technique. In this study the sampling used was non-probability sampling, in which the researcher only determines respondents who are sampled for research based on certain criteria (the respondents must only have authority as regional government officials) and the number of samples was determined from the sub-district in Bulukumba Regency. Regarding the number of population and samples, it can be seen in Table 1 as follows:

**Table 1.** Number of Respondents by Subdistrict in Bulukumba District.

| Number | Sub-District | Area (Km$^2$) | Total Population | Number of District Employees (Population) | Total Sample |
|--------|--------------|---------------|------------------|-------------------------------------------|--------------|
| 1 | Gantarang | 173.51 | 75,549 | 88 | 22 |
| 2 | Ujung Bulu | 14.44 | 55,615 | 64 | 16 |
| 3 | Ujung Loe | 144.31 | 41,921 | 52 | 13 |
| 4 | Bonto Bahari | 108.60 | 25,594 | 32 | 8 |
| 5 | Bontotiro | 78.34 | 21,575 | 28 | 7 |
| 6 | Herlang | 68.78 | 24,639 | 28 | 7 |
| 7 | Kajang | 129.06 | 49,032 | 48 | 12 |
| 8 | Bulukumpa | 171.33 | 52,599 | 58 | 14 |
| 9 | Rilau Ale | 117.53 | 40,339 | 48 | 12 |
| 10 | Kindang | 148.76 | 31,463 | 36 | 9 |
| | Total | | | 482 | 120 |

Source: Reference [29].

### 3.3. Data Analysis

The results of the identification and analysis determine leading sectors in Bulukumba Regency, as a strategy in the regional economic development of Bulukumba Regency. To answer the hypotheses that have been formulated, several data analysis methods were used; namely Location Quotient analysis is used to determine the base and non-base sectors in theBulukumba Regency economy, shift-share analysis was used to determine changes and shifts in the economic sectors of the Bulukumba Regency region, multiple linear regression analysis to determine the effect of independent variables on the dependent variable, and SWOT analysis to determine economic development and investment strategies in the Bulukumba Regency area.

### 3.3.1. Location Quotient Analysis (LQ)

The LQ method is one of the approaches commonly used in the basic economic model as a first step to understanding the activity sector of the Gross Regional Domestic Product (GRDP) of Bulukumba Regency as a growth driver [29].

$$LQ = \frac{GRDP\ bi / \sum GRDP\ b}{GRDP\ ssi / \sum GRDP\ ss} \tag{1}$$

PDRBbI is PDRB sector i in Bulukumba Regency in a certain year; ∑PDRBb is total GRDP in Bulukumba Regency in a certain year; PDRBssi is PDRB sector i in South Sulawesi province in a certain year; ∑PDRBss is Total GRDP in South Sulawesi province in a certain year.

Based on the formulation shown in the above equation, three possible LQ values can be obtained [61]:

(1)　LQ value = 1, this means that the level of specialization/basis of sector i in Bulukumba Regency is the same as the same sector in the economy of the province of South Sulawesi.
(2)　LQ value > 1, this means that the level of specialization/basis of sector i in Bulukumba Regency is greater than the same sector in the economy of the province of South Sulawesi.
(3)　LQ value < 1, this means that the level of specialization/basis of sector i in Bulukumba Regency is smaller than the same sector of the economy of the province of South Sulawesi.

### 3.3.2. Shift-Share Analysis

The results of the shift-share analysis will illustrate the sector performance in the Gross Regional Domestic Product (GRDP) of the Regency of Bulukumba compared to the sector performance in the province of South Sulawesi. The deviations were analyzed, which determined the results of the comparison. If the deviation is positive, it is said that a sector in the Gross Regional Domestic Product (GRDP) of Bulukumba Regency has a competitive advantage or vice versa the region. Components of the shift-share analysis include the provincial share, proportional shift and differential shift. The formulations of each of the three components are described as follows [59,62]:

$$PEK = KPN + KPP + KPK \tag{2}$$

The indicators assessed are PEK is the change in district income, KPN is a component of provincial growth, KPP is a component of proportional growth, and KPK is a component of regency competitiveness growth.

### 3.3.3. Multiple Linear Regression Analysis

The analytic method used to see the effect between variables is multiple linear regression, performed with the SPSS software program [63,64]. This is used to prove the hypothesis proposed, the general form of multiple linear regression models is as in the following equation:

$$PE = \beta o + \beta 1 SDA + \beta 2 HR + \beta 3 KB + \beta 4 RE + \varepsilon \tag{3}$$

PE is economic growth; SDA is natural resources; HR is the competence of human resources; KB is a community culture; RE is regulation. $\beta 1, \beta 2, \beta 3, \beta 4$ = Regression coefficient; $\beta o$ = Constant. Statistical testing of the model used a t-test (partial test) and an F-test (simultaneous test). The *t*-test can be detected by looking at the number of degrees of freedom or (df) and the degree of confidence of 5%. Then, Ho (the hypothesis that an independent variable individually affects the dependent variable) can be rejected or accepted, and the coefficient of determination $R^2$ can be tested. If the F value is greater than the F-table, then Ho can either be rejected at a 5% confidence level or the hypothesis that

all independent variables simultaneously and significantly influence the dependent variable can be accepted. The coefficient of determination test ($R^2$) can be detected by looking at the adjusted $R^2$ value.

### 3.3.4. SWOT Analysis

SWOT consists of analyzing strengths, weaknesses, opportunities and threats. This analysis is a strategic planning technique. The main advantage of SWOT analysis is its simplicity, which has resulted in its continued use in both leading companies and academic communities [65]. Depending on cell location, the attributes related to the organization's product or service are considered to be major or minor strengths or weaknesses described in Table 2 [66,67]. For more details, it can be seen in Table 2 as follows:

**Table 2.** Matrix of SWOT Analysis.

| Internal Eksternal | Strength | Weakness |
|---|---|---|
| Opportunity | Strategy SO<br>Strategies that utilize the ability to get opportunities in activities | Strategy WO<br>Strategies that reduce existing weaknesses by exploiting opportunities that can provide benefits |
| Threats | Strategy ST<br>Strategies that use force to the maximum in overcoming all threats | Strategy WT<br>Strategies that minimize weaknesses to take advantage of opportunities |

Source: Reference [66].

## 4. Results

### 4.1. The Determination of Natural Resource Factors on Economic Growth in Bulukumba Regency

The economic growth rate of Bulukumba Regency in 2018 was 5.05%, with the value of Gross Regional Domestic Product (GRDP) in 2018 reaching 8120.98 billion rupiahs. If the GRDP growth is ranked according to the economic category the highest is the mining and quarrying category at 12.85% and the lowest is the electricity and gas procurement category at 1.67%. The largest Gross Regional Domestic Product (GRDP) distribution in the agriculture category was 43.07%, followed by 14.76% in the category of wholesale and retail trade, car and motorcycle repair. The third highest construction category is around 8.90%, 7.00% in the industry category, and 6.87% in the government administration category. This shows the dominance of the agricultural sector which provides the largest contribution to the distribution of GRDP, however, it is not yet fully developed and optimally large because the position of Bulukumba Regency is ranked 20 in the South Sulawesi region in terms of per capita Gross Regional Domestic Product (GRDP) differences. This shows the problems faced by Bulukumba Regency in the future in terms of increasing economic potential.

### 4.1.1. Agricultural and Plantation Potentials

The agricultural sector still plays an important role in economic development in Indonesia. Bulukumba Regency puts the agriculture sector as one of the leading sectors that provide the greatest contribution to the economy of the Regency. This is supported by extensive land resources, a suitable climate and great genetic diversity of biological resources. This condition is reflected in the vast agricultural potential which consists of paddy fields, among other land types, in 201,972,878.9 Ha with production reaching 370,713.00 tons. The area in Bulukumba Regency is divided into sub-districts, and the Gantarang sub-district is an area that has agricultural production, and has an area of 16,429 hectares of agricultural land and agricultural production produced in 2019 reached 65,105 tons. Graphically, the agricultural potential can be seen in Figure 3 as follows:

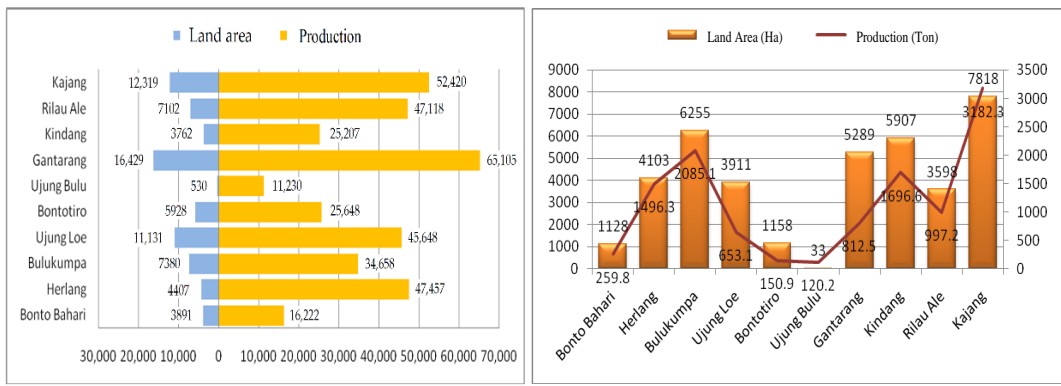

**Figure 3.** Agricultural and Plantation Production by District in Bulukumba Regency. (Source: Reference [29]).

Potential plantation crops also include superior crops such as deep coconut, hybrid coconut, robusta coffee, cocoa, clove, cashew, rubber, cotton and pepper. Plantations are necessary all activities that commercialize certain plants on the soil and/or other growing media in the appropriate ecosystem. Many areas of Bulukumba Regency are highlands with cold temperatures, so that plantation crops are suitable to grow in the region. Data released by BPS Statistics of Bulukumba illustrate that the Kajang sub-district is a mountainous region so that plantation crops reach 7818 ha with production reaching 3182 tons. Likewise, the Bulukumpa sub-district region is an area that grows a lot of plantation crops such as cloves, coconuts and other plants, and the area of plantations reaches 6255 ha with production reaching 2085 tons, while the area of Kindang sub-district is also one of the areas in Bulukumba district producing plantation crops with plantation land area reaching 5907 ha and with production in 2018 reaching 1696 tons. Plantation plants that grow widely in Bulukumba Regency are plants that have high economic value such as cloves, rubber, coffee, cocoa and forest wood. Figure 4 shows the growth in the plantation sector, namely the growth rate of the plantation sector including coconut plantations (9.7%), rubber (7.66%), coffee (5.32%), cocoa (17.32%), cloves (1.83%), pepper (0.90%) and forest products (57.28%). The figure gives the impression that Bulukumba Regency has the potential for natural resources in a very large plantation sector but that it has not been managed optimally.

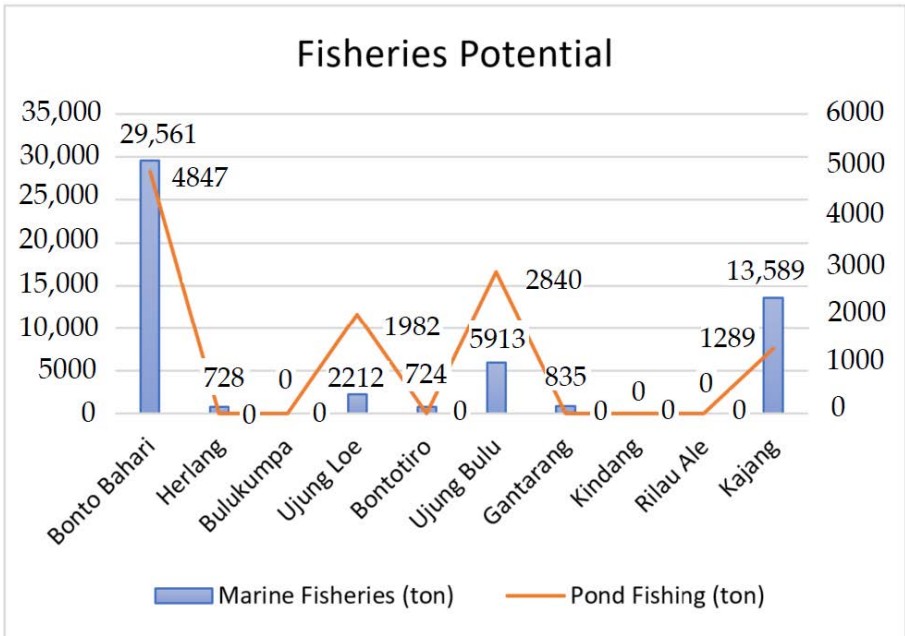

**Figure 4.** Production of Fisheries Products in Bulukumba Regency. Source: Reference [29].

### 4.1.2. Fisheries Potential

Fisheries are necessary for activities related to the management and utilization of pre-production, production, processing and marketing carried out in a fisheries system. Fisheries statistics are divided into data on capture fisheries and aquaculture. Of the ten sub-districts in Bulukumba Regency that have the potential for fisheries, especially capture fisheries, there are only seven sub-districts which have fisheries, because these areas are by the sea, so many people engage in economic activities in the fisheries sector (both fishing and marine fisheries).

Data in 2018 showed Bulukumba Regency's fisheries production reached 53,562 tons, or 83% of overall fishery production, and for aquaculture reached 10,958 tons, or 17%. The potential of aquaculture is very large when viewed from the position of the area surrounded by seas and rivers, but the contribution is still small to the regional economy. To be able to contribute specifically to regional income for the marine and fisheries sector, it must be able to be utilized optimally; if this is lacking, it will result in a large amount of aquaculture land available in several sub-districts not being used optimally.

### 4.1.3. Industry and Tourism Potential

Industrial allotment areas in Bulukumba Regency consist of three types of allotment areas; namely, large industrial allotment areas which are designated as Gantarang sub-district areas with cotton and wood processing industries, Ujung Loe sub-district, and Bulukumpa sub-district with a rubber processing industry. Medium industrial allotment areas include a Bontobahari sub-district area with a shipbuilding industry, while the household industry allotment area in the form of an agglomeration of home industry areas is set in several Kajang, Ujung Loe, Ujung Bulu, Bontotiro, Herlang, Kindang, Rilau Ale areas and parts of the Bulukumpa sub-district.

Figure 5 shows the development of the industrial sector. The food industry had the highest growth rate, which had production growth that reached 26.9%. The clothing industry contributed 19.6%, the chemical industry contributed 21.7%, the metal industry 11.7% and the handicraft industry was quite developed in Bulukumba Regency and contributed 20.01%. This potential needs to be managed properly so that it can contribute to the economy and prepare employees for the community.

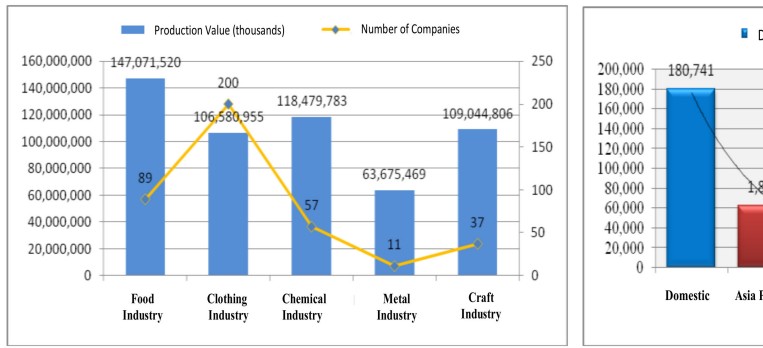 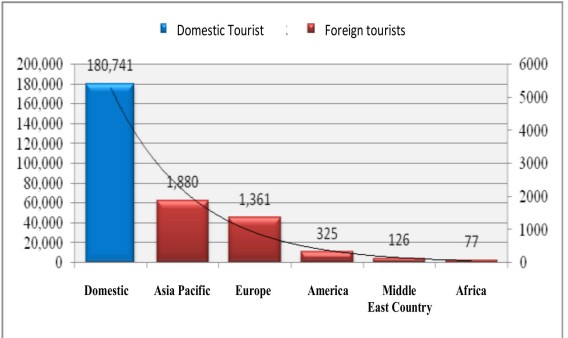

**Figure 5.** Total Production of Industrial and Tourist Products in Bulukumba Regency. Source: Reference [29].

The data has shown that Bulukumba Regency, as a potential tourist destination area in the province of South Sulawesi, is an area with increasing tourist arrivals from year to year. Bulukumba Regency has the potential to attract tourism in the form of nature tourism, cultural tourism and marine tourism which have different characteristics and specifications. Some tourism products are superior in the area that is not present in other regions such as the craft of "Phinisi" boat making as a product of the community culture and the culture of "amma toa" in Kajang that have not been well managed [41]. Bulukumba Regency could serve as an attraction to be visited by both domestic and foreign tourists.

All these kinds of tourism in Bulukumba Regency have potential, however, in terms of management they have not contributed much to regional revenue. Management of various strategies needs to be done so that tourists want to come to visit the tourist area.

Figure 5 shows the development of tourist arrivals in Bulukumba Regency in 2018. Most tourists were domestic tourists, reaching 180,741 people or 97% of all tourists, while foreign tourists who came to Bulukumba Regency made up only 3%. We conclude that there are still very few foreign tourists visiting tourist sites in Bulukumba. Foreign tourists could contribute a lot to the region as a contributor to foreign exchange. To achieve this, various improvements need to be made to facilities and roads and supporting facilities need to be developed related to tourism activities such as cultural arts performances that have regional characteristics.

### 4.1.4. Location Quotient Analysis and Shift-Share

There is so much economic potential in Bulukumba Regency, but it is also necessary to understand existing innovations in the region. It seems, therefore, that constructed advantages based on regional innovation systems that transmit over long distances as well as regional networks are becoming the models of choice for achieving regional economic development [68]. The level of innovation is one of the parameters in the concept of sustainable cities, the higher the level of competitiveness of an area, the higher the level of community welfare. LQ value can be used as a guide to determine potential sectors to be developed, because a sector cannot only meet the needs from within the region, but can also meet the needs in other regions or generate surpluses. Based on the results of the LQ analysis the leading sectors which have potential in Bulukumba Regency based on the sub-district level are as follows:

From the results of the calculation of the Location Quotient GRDP index of Bulukumba Regency during the 2019 monitoring year, the base and non-base sectors can be identified. An LQ value > 1 means a sector is more dominant than the sector at the provincial level and indicates that the Regency surplus will be the product of that sector. Assuming an LQ value < 1 means that the role of the sector is smaller in the District than it is at the provincial level.

Based on the LQ index calculation results, Table 3 shows the average by sector in Bulukumba Regency, namely the agriculture sector and the non-agriculture sectors. This indicates that Bulukumba Regency has been able to meet its own needs and make it possible to facilitate the outside of the area of goods and services. The Agriculture Sector is the sector with the highest LQ value, which is an average of all agricultural products (1032) followed by the fisheries sector with an LQ value of 1012. Although the base sector is a sector that has the potential to be developed and can spur economic growth in Bulukumba Regency, the role of the non-base sector cannot be avoided, because the base sector will be able to help the development of the non-base sector into a new base sector. For example, base sectors that are limited by the maximum of natural resources of Bulukumba Regency by presenting investors to manage resources that have potential that has not yet been managed.

**Table 3.** Results of LQ Analysis of Potential Sector in Bulukumba Regency.

| | Economic Sector | Value of LQ | Description |
|---|---|---|---|
| | **A. Food Crop Agriculture** | | |
| 1. | Paddy | 1.2 | Leading sectors |
| 2. | Corn | 0.9 | Non Basis |
| 3. | Soy | 1.3 | Leading sectors |
| 4. | Peanuts | 1.1 | Leading sectors |
| 5. | Green beans | 0.6 | Non Basis |
| 6. | Cassava | 1.0 | Leading sectors |
| 7. | Sweet potato | 0.9 | Non Basis |
| 8. | Mango | 1.1 | Leading sectors |
| 9. | Durian | 0.8 | Non Basis |

**Table 3.** *Cont.*

| | Economic Sector | Value of LQ | Description |
|---|---|---|---|
| 10. | Orange | 1.6 | Leading sectors |
| 11. | Banana | 1.3 | Leading sectors |
| 12. | Papaya | 1.5 | Leading sectors |
| 13. | Pineapple | 0.7 | Non Basis |
| 14. | Shallot | 0.7 | Non Basis |
| 15. | Chili | 1.0 | Leading sectors |
| 16. | Petain | 0.9 | Non Basis |
| **B. Plantation** | | | |
| 1. | Coconut | 0.8 | Non Basis |
| 2. | Rubber | 0.9 | Non Basis |
| 3. | Coffee | 0.6 | Non Basis |
| 4. | Cocoa | 1.3 | Leading sectors |
| 5. | Clove | 0.2 | Non Basis |
| 6. | Pepper | 0.5 | Non Basis |
| 7. | Plywood | 1.5 | Leading sectors |
| **C. Fishery** | | | |
| 1. | Catch fisheries | 1.0 | Leading sectors |
| 2. | Seaweed Cultivation | 1.3 | Leading sectors |
| 3. | Freshwater aquaculture | 1.8 | Leading sectors |
| **D. Industry** | | | |
| 1. | Food industry | 1.5 | Leading sectors |
| 2. | Clothing industry | 1.1 | Leading sectors |
| 3. | Chemical Industry and Building Materials | 0.9 | Non Basis |
| 4. | Metal and Electronic Industry | 0.4 | Non Basis |
| 5. | Handicraft industry | 0.6 | Non Basis |
| **E. Tourism** | | | |
| 1. | Means of Accommodation (Hotels) | 0.6 | Non Basis |
| 2. | Type of Tourism Business (Travel) | 0.5 | Non Basis |
| 3. | Tourism Support Business (Resto and souvenir shop) | 0.9 | Non Basis |

Source: Analysis Results, 2020.

### 4.1.5. Shift-Share Analysis

Bulukumba Regency GRDP data was analyzed using shift-share analysis to determine which sectors experienced rapid growth and which sectors that grew slowly. The analysis was determined by determining the value of the Provincial Growth Component (KPN), Components of Proportional Growth (KPP) and Components of Regency Competitiveness Growth (KPK).

Table 4 explains that KPN is Provincial Growth Component, KPK is Components of Regency Competitiveness Growth, KPP is Components of Proportional Growth and PN is Changes in Regency Revenues. The analysis shows that several sectors have an allocation effect or have the potential to contribute to the Gross Regional Domestic Product (GRDP) in Bulukumba Regency. The contribution

to economic growth can be seen in Table 4, based on the shift-share component divided into four categories; namely (i) shifts in the economic structure of a region influenced by national or KPN with a value of 541.16, (ii) proportional shifts based on the gross value of each sector against the total sector of provincial economic activity with a KPP value of 4.59 or PP > 0. This means that the sector of economic activity is growing rapidly whereas a PP < 0 means it is not a specialization of the provincial economic activity sector, (iii) differential shift or competitive position based on gross value added in the same sector as the KPK of 178.22 or PPK > 0, for details, it can be seen in Table 4 as follows:

**Table 4.** Changes in Economic Growth in the District of Bulukumba.

| No | Economic Sector | Bulukumba Regency PDRB | | | |
|---|---|---|---|---|---|
| | | Absolute Value of Bulukumba Regency | | | |
| | | KPN | KPP | KPK | PEK |
| 1. | Agriculture, Forestry & Fisheries | 224.26 | −53.81 | 168.89 | 1.55 |
| 2. | Mining & Excavation | 11.04 | −9.39 | 18.60 | 20.26 |
| 3. | Manufacturer | 35.78 | −30.96 | 15.41 | 20.23 |
| 4. | Electric & Gas | 0.96 | 0.03 | −0.04 | 0.95 |
| 5. | Water Supply, Waste Disposal, Waste Management & Remediation Activities | 0.22 | −0.01 | −0.01 | 0.20 |
| 6. | Construction | 47.01 | 10.72 | 6.67 | 64.42 |
| 7. | Wholesale & Retail Trade; Motorcycle & Motorcycle Repair | 83.83 | 54.70 | −22.06 | 116.47 |
| 8. | Transportation & Storage | 11.61 | 5.51 | −1.42 | 15.69 |
| 9. | Accommodation & food service activities | 3.35 | 2.73 | −1.95 | 8.04 |
| 10. | Information & Communication | 20.31 | 14.48 | −7.88 | 26.90 |
| 11. | Financial Activities & Insurance | 18.43 | −6.13 | 1.84 | 14.14 |
| 12. | Real Estate Activity | 25.23 | −8.53 | 8.12 | 24.82 |
| 13. | Business activities | 0.10 | 0.04 | 0.01 | 0.15 |
| 14. | Public & Defense Administration; Mandatory Social Security | 32.83 | 13.89 | −7.34 | 39.38 |
| 15. | Education | 4:30 | 6.45 | −1.50 | 21.24 |
| 16. | Human Health and Social Work Activities | 5.86 | 1.33 | 0.89 | 8.09 |
| 17. | Other Service Activities | 4.04 | 3.54 | −0.01 | 7.57 |

Source: Analysis Results.

The results of the shift-share indicate that agriculture, mining, quarrying, processing, construction, financial services, corporate services, and health services are competitiveness when compared to other sectors of economic activity. The average value of PEK was 22.94, which means that Bulukumba Regency has an economic potential to develop going forward because it is supported by the potential of many available natural resources that have not been managed. A national development approach that greatly emphasizes macroeconomic growth tends to ignore the great inter-regional development inequality that exists [69]. Furthermore, the shift from a traditional economy to a circular economy requires the realization of environmental innovations and sustainable engineering solutions [70].

## 4.2. The Influence of Natural Resources, Human Resources, Community Culture, and Regulation on Regional Economic Growth

Economic growth is one of the macro indicators used to determine the real economic performance of a region. The rate of economic growth is calculated based on changes in Gross Regional Domestic Product (GRDP) based on the constant price of the relevant year. Economic growth can be seen as an increase in the number of goods and services produced by all business fields of economic activity in an area over a year. To determine the economic growth in an area, it is necessary to know the factors that

influence it. For Bulukumba Regency, it is determined that an important factor influencing economic growth is the influence of potential natural resources [71]. The available natural resources can improve the regional economy in the form of contributions to regional income. The second factor is the factor of human resources; natural resources available without human resources will not advance the economy. The third factor is culture, which is also very decisive in driving economic growth. Determining economic growth is important because the debate on the nature and dynamics of regional development in both academic and policy circles has now moved on from the earlier focus on endogenous regional assets such as localized networks of association and trust to analyze the complex relationship between economic globalization and regional change [72,73].

The magnitude of these factors on the regency's economic growth can be determined by the statistical means in the form of Multiple Linear Regression with the SPSS program, the results of data processing using the program can be seen in Table 5 as follows:

**Table 5.** Summary of Statistical Results of Multiple Regression Analysis.

| Variable Regression | Regression Coefficient | T-Count | T-Table | A Constant | Sig |
|---|---|---|---|---|---|
| | | | | 1559 | 0.000 |
| SDA | 705 | 9662 | 1645 | | 0.000 |
| HR | 130 | 2195 | 1645 | | 0.030 |
| KB | 0.250 | 1357 | 1645 | | 0.178 |
| RE | 0.212 | 1048 | 1645 | | 0.297 |
| R-square =0.472 | | | | | |
| F-count = 25.718 | | | | | |
| F-table = 2680 | | | | | |

Source: analysis results.

Table 4 shows the results of multiple linear regression calculations. PE is economic growth (the dependent variable), SDA is natural resources (independent variable), HR is human resources, KB is the culture of the society, and RE is a regulation. Based on calculations with the help of SPSS Software using the Full Model Regression, the regression equation was obtained as follows:

$$PE = 1.559 + 0.705\ SDA + 0.130\ HR + 0.250\ KB + 0.212RE$$

The regression equation has a value of βo or a constant value of 1.559. This shows that if the independent variable is considered cash, then the effect on economic growth is 1.559. The coefficient value of the positive independent variable means that if there is a change it will cause a direct change in the dependent variable or the PE variable (economic growth). The natural resource coefficient (natural resource) is 0.705 which means that if natural resource production rises by one unit, it will affect the regional economic growth increase by 0.705 units, assuming that other variables are considered constant. The HR regression coefficient (human resources) rose by one unit, which will affect regional economic growth by 0.130 units, assuming that other variables are considered constant. The human resources regression coefficient rose by one unit, affecting the economic growth of the region by 0130 units, assuming that other variables are considered constant. The coefficient of KB (culture) rose by one unit, will affect the economic growth of the region (PE) by 0250 units, assuming that the other variables are considered constant, and the RE (regulation) value rose by one unit, and will affect the economic growth by 0212 units.

The F statistical test, or simultaneous significance test, basically shows whether all independent variables included in the model have an influence together on the dependent variable. This F test is done by comparing the F-count with the F-table value at the real level $\alpha = 0.05$. The F test has a significant effect if the F-count is greater than the F-table or the error probability is less than 5% ($p < 0.05$). From the calculation results of the full model regression analysis with the help of the SPSS program, the F-count is 30,038 with a probability level of 0.000 (significant). While the F-table is 2680. Thus,

the F-count is greater than the F-table (25,718 > 2680), and also the probability is much smaller than 0.05. This means that the SDA, SDM, KB and RE together affect the economic growth of the Bulukumba Regency. The magnitude of the influence (contribution) of the independent variables together on the dependent variable can be seen from the magnitude of the double determinant coefficient ($R^2$). The coefficient of determination is between zero and one. If $R^2$ obtained from the calculation results is increasing (close to 1), then it can be said that the influence of the independent variable on the dependent variable is increasing. The result of the coefficient of determination is 0.472 or 47.2% (the effect of the independent variable on the dependent variable). The coefficient of determination is between zero and one. If $R^2$ obtained from the calculation results is increasing (close to 1), then it can be said that the influence of the independent variable on the dependent variable is increasing.

$T$-tests are used to test the significance or lack thereof of partial regression coefficients. Testing through $t$-test is sone by comparing the t-count value with a t-table value at the real level $\alpha = 0.05$. The $t$-test has a significant effect if the calculation result of t-calculation is greater than the t-table (t-count > t-table) or the probability of error is less than 5% ($p < 0.05$). Independent variable influence was determined based on the results of statistical calculations. The natural resource variable (SDA) had t-count = 9662 and t-Table = 1.645; the t-count value is greater than the t-table value (9662 > 1645) which means that $H_0$ is rejected, meaning the natural resource variable influences the economic growth of Bulukumba Regency. The human resources variable (HR) had t-count = 2.195 and t-table = 1.645; the t-count value is greater than the t-table value (2.195 > 1.645) which means that $H_0$ is rejected, which means that HR affects the economic growth of Bulukumba Regency. The KB variable (culture) had t-count = 1.357 and t-table = 1.645; the t-count value is smaller than the t-table value (1.357 < 1.645), which means that $H_0$ is accepted, meaning that it is not significant. In this case, culture has not had a large effect on economic growth in Bulukumba Regency, while for the RE (regulation) variable t-value = 1.048 and t-table = 1.645; the t-count value is smaller than the t-table value (1.048 < 1.645), so $H_0$ is accepted meaning the variable regulation is insignificant. Regulation for economic growth in Bulukumba Regency is still lacking.

### 4.3. Analysis of Investment Opportunities in Bulukumba Regency

An analysis determining leading sectors is needed as a basis for formulating economic development policy patterns in Bulukumba Regency in the future so that economic development policies can be directed to drive the sector. The priority of economic development in Bulukumba Regency must be based on the superior sector. It must not be solely based on available natural resources but must also pay attention to technological resources, human resources and culture so that the resulting output will be competitive and support the specific potential of the region. To further develop the region, it is necessary to engage in direct development such as marketing development, destination development, partnership development and territorial arrangement to spur development and economic growth of the region, thereby encouraging community welfare.

The investment program strategy in Bulukumba Regency uses, as a reference for investment, the potential and characteristics of Bulukumba Regency, and by considering internal factors and external factors, is able to contribute to the economic development of Bulukumba Regency and have an impact on the welfare of the community. The following is a SWOT table that shows potential investment improvement strategies that can be developed in Bulukumba Regency with the aim of sustainable development in the region. The SWOT results can be seen in Table 6 as follows:

Based on the results of a SWOT analysis of increased investment in Bulukumba District, this strategy uses internal strength to take advantage of external opportunities, by maximizing the existing power to benefit and by achieving sustainable economic growth. Investment requires various strategies to develop, and these that cannot be separated from the opportunity to follow up on issues and problems, so that desired economic growth and improvement of community welfare in the area of the Regency of Bulukumba can be achieved properly. A measurable SWOT factor enables an organization to prioritize SWOT factors in creating an action plan, and facilitates an organization's ability to efficiently formulate

strategic planning for maintaining or enhancing customer satisfaction, thereby gaining a competitive advantage [74].

**Table 6.** SWOT Matrix of Investment Improvement Strategies in Bulukumba District.

| | Strength | Weakness |
|---|---|---|
| **Internal External** | 1. Economic potential<br>2. South-south strategic location<br>3. Number of workers<br>4. Culture | 1. Investment regulations are still lacking<br>2. Infrastructure is still limited<br>3. Low quality of human resources<br>4. Culture is still low on economic activity |
| **Opportunity** | **Strategy (SO)** | **Strategy (WO)** |
| 1. Demand for agricultural products increases<br>2. Market share is open for export<br>3. The development of technology is quite rapid | 1. Increased production and development of superior quality<br>2. Increasing investor interest in managing the development of agribusiness and tourism businesses in the Bulukumba region.<br>3. Utilization of technology for superior economic production | 1. Legal certainty and ease of investment<br>2. Improve, develop, and develop infrastructure in basic infrastructure and trade in a sustainable manner.<br>3. Developing the potential of regional-based productive community economic activities and local wisdom. |
| **Threats** | **Strategy (ST)** | **Strategy (WT)** |
| 1. Agricultural production in other regions is higher<br>2. More investors in other regions<br>3. High credit interest rates | 1. Increase the competitiveness of local products through increased local production, the productivity of economic activities<br>2. Optimizing the marketing and economic potential of Bulukumba Regency<br>3. Empowerment of micro, small, medium, and cooperative businesses. | 1. Optimizing the investment license service system<br>2. Improve technology-based human resources skills<br>3. Provision of investment facilities, facilities, and/or incentives. |

Source: analysis results.

## 5. Discussion: Open Innovation Dynamics and Regional Development

### 5.1. The Effects of Economic Potential

An analysis which determines leading sectors is needed as a basis for the formulation of future economic development policy patterns in Bulukumba Regency, so that economic development policies can be directed to drive the sector. Regional economic development priorities in Bulukumba Regency must be based on leading sectors and not only based on natural resources owned. However, they must also focus on technology and the quality of human resources so that the resulting output will be highly competitive, because it is supported by the potential of the region. A lock-in functions like a resonance which transforms the resonating dynamics. It cannot be known ex-ante which dimensions in the

multidimensional arrangement of industry, academia, and governance will be able to retain wealth from the incursive transformation [68,75,76].

Based on the results of the analysis using LQ and shift-share, and the results of the regression that showed a significant of 0.00 < 0.05, Bulukumba Regency has huge natural resource potential; making the agriculture, mining, quarrying, construction, financial services, corporate services, and health services industries competitive would provide the greatest contribution to the economy of the Regency of Bulukumba. The magnitude of this potential is supported by vast land resources, suitable climate, and diversity in all regions or districts in Bulukumba Regency. The economic sector as a base sector in Bulukumba Regency is still very limited, meaning that there is still a lack of marketing outside the region of products produced. Currently the economic sector is only used for fulfillment within the region, and there is further need of improvement in regulation of economic activity. Regional economic stability needs to be maintained so that businesses and the public can make various efforts in a planned manner. Economic stability includes low inflation and clear business regulations accompanied by consistent law enforcement and no security disturbances [77,78].

Economic development activities that exclusively look at the leading sectors may have no effect apart from having an impact on the acceleration of economic growth. The modern economy can be modeled using the Dynamics of Entrepreneurial Cycle of Open Innovation with three sub-economies, such as open market innovation by SMEs and start-ups, closed open innovation by big business and social open innovation [79]. Leading sectors are sectors or economic activities that have potential and prospects by means of fostering an entrepreneurial spirit that allows them to drive economic business activities and create independence in regional development.

## 5.2. Influence of Human Ressources and Culture

The independence of regional development cannot be separated from human resources as the driving force of the economy. This human resource factor has presented a new thought process in the study of economic development theories, which places it at the main axis of economic development on a global, national, and regional scale. Economic development strategies based on human resource development are considered very relevant and compatible with the conditions and character of economic development, especially in developing countries. Our statistical analysis in the form of a regression obtained a significance value of 0.036 < 0.05, which can be interpreted to mean that human resources have an influence on economic growth in Bulukumba Regency, but that the influence is still very small, meaning that the process of labor production that produces goods and services has not used much technology that produces quality and efficient work.

Human resource development must be the main foundation for economic development policies in the region, especially in the Regency of Bulukumba. Bulukumba Regency will continue lagging behind other regions in levels of economic prosperity such as quality and standard of living unless there is a very significant increase in human resource development. Therefore, in the process of regional economic development, in the context of reducing development disparities between regions, there must be an improvement in the quality of human resources. Improved quality of human resources can be achieved through the process of competency development and by increasing expertise through training related to job performance. Development is very important, particularly due to the continued emergence of changes related to technological progress. To tackle the long term challenges related to human capital, it is necessary to boost labor productivity and increase wages by reducing the groundless differentiation of the population's income [80].

The next discussion in this research is about the relationship of economic growth in Bulukumba Regency and the culture of the community. Culture is one of the driving factors in advancing the economy in a region. Our statistical analysis results obtained a significant value of 0.115 > 0.05, which means that the culture of the people of Bulukumba does not influence economic growth. In this case, many people have not used technology in the production process, more specifically in the

agricultural sector that is still using traditional methods, so production is still very low when compared to other regions in South Sulawesi.

In order to increase the community's contribution to the economic growth of Bulukumba Regency, it is necessary to change the mindset of the population or change the cultural attitude towards the economy by providing an understanding of innovation, and by using technology in the production process so that culture does not become an obstacle in the process of economic development. The development of a culture for the dynamics of open innovation is also important in the realization of true open innovation in public organizations, because it must overcome cultural barriers in addition to legal and institutional barriers [81]. Developed countries are developing economies and producing a lot due to technology-based innovation, especially in the current era (namely the era of the industrial revolution 4.0 (four zero point)), which means that all economic activities must develop technology-based innovation to increase the productivity of all activities the economy.

### 5.3. Investment and Development Strategy

Investment is an expenditure aimed at increasing or maintaining a stock of capital goods consisting of machinery, factories, offices, and other durable products that are used in the production process. Increased investment and a better allocation of capital across sectors may help start growth in countries that are well endowed in terms of natural resources. High-quality investment may ensure growth and enhance welfare in the presence of abundant natural resources [7].

The role of investment in an area is very important for economic growth. Bulukumba Regency's economy is still low due to the crisis human resources, which are still inadequate, and culture that has not been supportive. This encourages the regional government to look for sources of development funding both from within and from abroad. The development of investment in the region has not yet developed because the natural wealth of the region is abundant but not utilized properly. In fact, if the regional government used natural resources, they could do business which could increase regional income by maximally utilizing natural resources.

Considering the investment needed in order to increase growth and sustainable development, the investment strategy in Bulukumba Regency should involve looking at the potential sources of innovation in Bulukumba Regency by considering the dynamics of micro and macro innovation, so as to contribute to economic development and have an impact on community welfare. Economic, Community and Environmental Sustainability in the Fourth Industrial Revolution aims to respond to the Fourth industrial revolution in open innovation and cyber physics, from manufacturing to service industries [82]. After considering these factors, it is necessary to formulate a comprehensive strategy through the formulation of priority strategies to increase investment. The main strategic formulations are (1) Development of new economic centers through the development of strategic sectors according to the carrying capacity of the environment and the superior potential of each sub-district (2) Development of strategic growth centers, among others by encouraging the distribution of investment based on area and zoning (3) Establishment of trade and service zones aimed at providing space for the development of the economic sector through trade business fields, traditional markets, wholesale markets, communication centers and modern shops (4) Provision of facilities and an increase in investment incentives that encourage economic growth.

In order to encourage investment, regional governments are required to be proactive in innovating and informing the public through various media. The existence of information that is fast, accurate, and up to date will assist investors in analyzing areas and making investment decisions. In order to attract new investment and encourage increased investment through the provision of investment incentives and facilities, the strategic policy emphasizes policy options to provide various investment facilities rather than providing incentives [78].

## 6. Conclusions

Bulukumba Regency is one of the areas in South Sulawesi which is based on the agricultural sector so that the implementation of economic development is largely determined by agricultural products. The available natural resources are a determining factor in regional development, meaning that people engaged in economic activities are mostly engaged in the agricultural sector. Regional economic development still relies on this sector and its potential has not been maximally managed. This is due to many factors, namely human resource competence, community culture and regulations in terms of natural resource management.

Human resources have not played a significant role, meaning that the level of knowledge is still low compared to other areas, especially in the agricultural sector, due to the lack of education from the community regarding modern agricultural technology. Another factor is that the existing culture in the community is still strong, and maintains traditional methods in the agricultural production process, still using traditional agricultural tools such as the use of human and animal labor in working on agricultural land. There is also a cultural habit of carrying out agricultural activities just for personal fulfillment, rather than for commercial purposes, so that added value is not obtained by the community. There also has not been much use of technological advances in the agricultural production process as a potential resource. Another factor is the lack of existing regulations to help develop regional potential, such as regulations for investment activities, which become an obstacle to economic growth.

Management of existing natural resources can provide added value to society if existing human resources increase their competence in the management of the agricultural sector by changing their mindset so that they are only fulfilling needs for production economically, and culture must change according to modern developments by using technology in the production process, so that productivity can be increased and the added value increased, which would result in an increase in regional economic growth.

In addition to increasing economic growth, a strategy is needed that can encourage increased investment in Bulukumba Regency, such as increasing the competitiveness of local products through increasing local production, and increasing investor interest with various regulations in the form of easy permits in the management and development of various businesses such as the development of agribusiness and regional tourism. Bulukumba can provide facilities, investment incentives, and new uses of technology to produce superior economic production. We hope that the government, together with the community of economic actors, can implement our strategy so that the economy can grow and develop and contribute to regional economic growth.

**Author Contributions:** Conceptualization, H.S. and B.S.; methodology, H.S., B.S. and D.M. software, H.S. and D.N.A.A., formal analysis, H.S., B.S. and D.M, data curation D.N.A.A. and D.M., writing—original draft preparation, H.S., D.N.A.A. and D.M., writing—review and editing, H.S., B.S., and D.M. supervision, H.S. and D.M. All authors have read and agreed to the published version of the manuscript.

**Funding:** This research received no external funding.

**Conflicts of Interest:** The authors declare no conflict of interest.

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
