# Peer review of "The Role of Natural and Human Resources on Economic Growth and Regional Development: With Discussion of Open Innovation Dynamics"

_2199-8531, doi:10.3390/joitmc6040103_

Round 1

Reviewer 1 Report

In the abstract, the purpose of the research and the methods used were defined. In the article itself, the section regarding methods is not clear. Instead of presenting theoretically the role of sample in the research, it was necessary to describe the characteristics of the sample and the way of its selection. There is no information which of the presented results comes from this survey? What questions did the questionnaire include? What data was obtained?

It is also unknown who were respondents? What does ‘economic actors’ mean? Table 1 shows the number of inhabitants and not economic actors. In the section Materials and Methods, reference should be made to the data presented in Table 1, which shows that the sample was composed of 120 respondents. What percentage of the total population is it? Is it sufficient to draw general conclusions?

There are technical shortcomings in the text, e.g. the unit is missing (Line 52), there is a dot instead of a comma (Line 59), no explanation of the abbreviation used (Line 88), incomprehensible sentence or sentences(?) (Lines 4-8). Figure 1 is illegible, some fields are missing text.

Summarizing, the research on development strategies and the factors that determine it are interesting, but the way they are presented is not very clear.

Author Response

 Dear Reviewer

I thank you very much for your input to improve my research paper, I hereby send the results of the improvements according to the suggestions and input from the reviewer, Once again I thank you, hopefully it will be as expected.

Best regards and stay healthy

Reviewer 2 Report

See the attached file for my comments

Author Response

Dear Reviewer

I thank you very much for your input to improve my research paper, I hereby send the results of the improvements according to the suggestions and input from the reviewer, Once again I thank you, hopefully it will be as expected.

Best regards and stay healthy

  1. Comment Reviewer:

Title. The title is too long, and it does not suggest that the paper is a case study. I suggest the authors propose an alternative title

Autor’s Revision:

Based on the author's suggestions regarding the title change, the author has made changes to the new title, namely: “Economic Growth and Regional Development Strategy for Bulukumba Regency, South Sulawesi, Indonesia”

  1. Comment Reviewer:

Objectives. The potential impact of the research should be more clearly stated, as well as the potential users in the introduction section.

Autor’s Revision:

The author has added the objectives to be achieved in this paper, namely: “The objectives of this study are (1). to determine the potential use of natural resources and human resources as a determinant of economic growth in the Bulukumba district, (2). to find out how the influence of natural resources, human resources, community culture and regulation on economic growth in Bulukumba district”.

4. In the introduction, because it is long, it has been reduced by the author, it can be seen in the revised manuscript. The author has reduced the conceptual framework according to the instructions from the reviewer

Comment Reviewer:

Please, try to reduce the length of Section 5 and 6 and focus on the main contribution of the analysis to the aim of the paper

Autor’s Revision:

The author has made a reduction in sections 5 and 6 regarding the results and discussion can be seen in the manuscript

  1. Comment Reviewer:

Autor’s Revision:

The methodology has been improved and sections 3 and 4 have been combined as follows:

3. Material and Methods

This research is a type of qualitative survey research that describes systematically, factually, and accurately a treatment in the area that is the object of research, while the quantitative method for testing the hypothesis consists of:

Hypothesis 1 (H1): The economic potential owned by Bulukumba District can increase the economic growth of the region.

Hypothesis 2 (H2): The influence of natural resources, human resource competencies and cultural culture, and regulatory factors in the region of the Bulukumba Regency can encourage economic growth with strategies according to regional characteristics.

This is done to reveal trends and to prove simple statistics various quantitative data. The philosophical reasons for combining both approaches were triangulation logic, qualitative research results were re-checked in the quantitative approach and vice versa

  1. Comment Reviewer:

Autor’s Revision:

The author has made improvements to the conclusion material according to the reviewer 's direction, namely by adding to the results of the conclusions and arranging them clearly based on existing factors, namely human resource competence and community culture, and regarding regulations, as follows: “Bulukumba Regency is one of the areas in South Sulawesi which is based on the agricultural sector so that the implementation of economic development is largely determined by agricultural products. The available natural resources are a determining factor in regional development, meaning that people in economic activities are mostly engaged in the agricultural sector from various sectors. So that regional economic development still relies on this sector, the potential has not been maximally managed. This is influenced by many factors, namely human resource competence, community culture and regulations in terms of natural resource management.

Human resources have not played much role, meaning that the level of knowledge is still low compared to other areas, especially in the agricultural sector, due to the lack of education from the community regarding modern agricultural technology. Another factor is that the existing cultural culture in the community is still strong, namely maintaining traditional methods in the agricultural production process, still using traditional agricultural tools such as the use of human and animal labor in working on agricultural land and there is also a culture of the habit of carrying out agricultural activities just for fulfillment. the need alone is not for commercial purposes so that added value is not obtained by the community, this also has not made much use of technological advances in the agricultural production process as a potential resource, another factor is the lack of existing regulations to develop regional potential such as regulations for investment activities, thus becoming an obstacle to economic growth

Management of existing natural resources can provide added value to society if existing human resources increase their competence in the management of the agricultural sector, namely changing the mindset from production to only fulfilling needs to production economically, as well as culture must change according to modern developments by using technology in the production process, so that productivity can be increased and the added value increases, which results in an increase in regional economic growth.

  1. Comment Reviewer:

References. Please, check carefully and select and homogeneous format for all the references

Comment Reviewer:

Autor’s Revision: The methodology has been improved and sections 3 and 4 have been merged

Round 2

Reviewer 1 Report

All my comments and suggestions were taken into account by the Author. The section Materials and Methods was supplemented with appropriate explanation. My other remarks were also included. In the new version of article I see only minor editorial shortcomings (grammar and punctuation).

Author Response

Dear Reviewer

Thank you for the suggestions and input for the improvement of my article, and it has been corrected based on suggestions from reviewers and editors, namely;
1. Regarding the title has been corrected according to the instructions from reviewers and editors
2. Changes have been made and added to point 5 regarding the discussion
3. At point 5.1. has been refined and added an explanation in conjunction with reference [81]: "Entrepreneurial cyclical dynamics of open innovation"
4. In point 5.2. has been added an explanation in accordance with reference [82]: "The culture for open innovation dynamics"
5. At point 5.3. has been refined and added an explanation with reference [83]: "Micro and macro dynamics of open innovation with Quatruple helix model"
Thus, this improvement has been made and can be seen in the manuscript I sent to reviewers and editors, hopefully it is in accordance with the expectations of reviewers and editors.

thank you

Reviewer 2 Report

The modifications you have made improved very much the quality of the paper. I consider that in the current form it can be published.

Author Response

Dear Reviewer

Thank you for the suggestions and input for the improvement of my article, and it has been corrected based on suggestions from reviewers and editors, namely;
1. Regarding the title has been corrected according to the instructions from reviewers and editors
2. Changes have been made and added to point 5 regarding the discussion
3. At point 5.1. has been refined and added an explanation in conjunction with reference [81]: "Entrepreneurial cyclical dynamics of open innovation"
4. In point 5.2. has been added an explanation in accordance with reference [82]: "The culture for open innovation dynamics"
5. At point 5.3. has been refined and added an explanation with reference [83]: "Micro and macro dynamics of open innovation with Quatruple helix model"
Thus, this improvement has been made and can be seen in the manuscript I sent to reviewers and editors, hopefully it is in accordance with the expectations of reviewers and editors.

thank you

This manuscript is a resubmission of an earlier submission. The following is a list of the peer review reports and author responses from that submission.